# Exploring the Association between Gut Microbiota and Inflammatory Skin Diseases: A Two-Sample Mendelian Randomization Analysis

**DOI:** 10.3390/microorganisms11102586

**Published:** 2023-10-19

**Authors:** Junhao Long, Jinglan Gu, Juexi Yang, Pu Chen, Yan Dai, Yun Lin, Ming Wu, Yan Wu

**Affiliations:** 1Department of Anesthesiology, Union Hospital, Tongji Medical College, Huazhong University of Science and Technology, Wuhan 430022, China; longjunhao1999@163.com (J.L.); yjxdoct@163.com (J.Y.); chen_pu2630@163.com (P.C.); 17866532506@163.com (Y.D.); franklinyun@hust.edu.cn (Y.L.); 2Institute of Anesthesia and Critical Care Medicine, Union Hospital, Tongji Medical College, Huazhong University of Science and Technology, Wuhan 430022, China; 3Key Laboratory of Anesthesiology and Resuscitation, Huazhong University of Science and Technology, Ministry of Education, Wuhan 430022, China; 4National Clinical Research Center for Child Health, Department of Nephrology, Children’s Hospital, Zhejiang University School of Medicine, Hangzhou 310058, China; gujinglan2000@163.com; 5School of Mathematics and Statistics, Central China Normal University, Wuhan 430079, China; 6Department of Dermatology, Union Hospital, Tongji Medical College, Huazhong University of Science and Technology, Wuhan 430022, China

**Keywords:** Mendelian randomization (MR), causality, intestinal flora, inflammatory skin diseases, genome-wide association studies (GWAS)

## Abstract

Emerging research underscores the substantial link between gut flora and various inflammatory skin diseases. We hypothesize that there exists a complex gut–skin axis, possibly affecting the progression of conditions such as eczema, acne, psoriasis, and rosacea. However, the precise nature of the causal connection between gut flora and skin diseases remains unestablished. In this study, we started by compiling summary data from genome-wide association studies (GWAS) featuring 211 unique gut microbiota and four types of skin conditions. We scrutinized these data across different taxonomic strata. Subsequently, we leveraged Mendelian randomization (MR) to ascertain if there is a causal link between gut microbiota and these skin conditions. We also performed a bidirectional MR analysis to identify the causality’s direction. By utilizing Mendelian randomization, we identified 26 causal connections between the gut microbiome and four recognized inflammatory skin conditions, including 9 positive and 17 negative causal directions. Additional sensitivity analyses of these results revealed no evidence of pleiotropy or heterogeneity. Our MR analysis suggests a causal connection between gut microbiota and skin diseases, potentially providing groundbreaking perspectives for future mechanistic and clinical studies on microbiota-affected skin conditions.

## 1. Introduction

Skin and subcutaneous disorders ranked as the 18th leading cause of disability-adjusted life years (DALYs) globally in the 2013 global burden of disease report, imposing an economic burden amounting to hundreds of millions of USD annually [1]. The human gut harbors a diverse array of microbial communities, with gut flora playing a pivotal role in human health. This includes modulating the immune system, impacting nutrient absorption, and influencing disease progression [2]. The gut–skin axis (GSA) is an emerging concept that delineates the interplay between the gut microbiota and the skin. The connection between gut flora and skin health, as well as disease, has gained considerable attention in recent research. The relationship between particular inflammatory skin conditions and the microbiome is believed to be influenced by an impaired intestinal barrier, enhanced inflammatory mediators, and byproducts discharged by microorganisms [3]. Short-chain fatty acids (SCFAs) are the main products of intestinal microbiota and act as a crucial energy provider for enterocytes. They play crucial roles in mucosal protection, immune regulation, and metabolism across various tissues [4]. SCFAs, encompassing propionate, acetate salts, and butyrate, are noted to restrain the growth, mobility, and adherence of inflammatory cells, thus manifesting anti-inflammatory impacts [5]. While the majority of related studies are still in their exploratory stages, there have been reports suggesting an association between intestinal flora and certain inflammatory skin diseases, such as eczema [6], acne [7], psoriasis [8], and rosacea [9]. Current research on gut flora and skin diseases may be constrained by confounding factors, including the environment, diet, and antibiotics [3]. In conclusion, the presence and orientation of a causal connection between intestinal microbiota and inflammatory skin conditions continue to be ambiguous. Consequently, it is essential to delve into the potential causal link between gut flora and inflammatory skin diseases.

Mendelian randomization is a robust method that integrates data from GWAS in a pooled manner. MR is a regularly used mechanism for extrapolating potential causal links between exposure elements and complex results. It capitalizes on genetic variants significantly correlated with exposure as instrumental variables to deduce causality [10]. Owing to the random allocation of genes at the moment of conception, they remain unaffected by confounding factors. This unique attribute of MR enables the reduction of confounding influences. If a genetic variant is linked with an exposure, and this exposure is subsequently linked with an outcome, it logically infers that the genetic variant should also maintain a connection with the outcome. In this study, we employed a two-sample MR to explore the potential causal link between the structure of the gut microbiome and the susceptibility to skin diseases as previously described. We focused on four skin conditions, namely, eczema, acne, psoriasis, and rosacea, using data from public GWAS repositories. By employing a bidirectional MR approach, we aimed not only to ascertain if the gut microbiota influences the risk of these dermatological diseases by chance but also to determine whether a genetic predisposition to these skin conditions has a causal effect on the gut microbiota. Our goal is to highlight the role of gut microbiota in the development of dermatoses. Such comprehension could eventually set the stage for the development of novel therapeutic strategies, encompassing probiotic therapy, dietary alterations, and fecal microbiota transplantation (FMT).

## 2. Method

### 2.1. Exposure Data

We picked single nucleotide polymorphisms (SNPs) linked with human gut microbial composition as instrumental variables (IVs) from the Genome-Wide Association Studies (GWAS) dataset of the international consortium MiBioGen [11]. This consortium carried out a large-scale, multiethnic GWAS encompassing 18,340 participants from 24 cohorts across the United States, Canada, and other countries. The study scrutinized microbial composition using gene sequencing profiles against 16S rRNA genes. It incorporated a total of 211 taxonomic units, consisting of 131 genera, 35 families, 20 orders, 16 classes, and 9 phyla.

### 2.2. Outcome Data

We procured dermatological data on eczema, acne, and psoriasis from FinnGen version 8 (https://r8.risteys.finngen.fi/, accessed on 15 May 2023) and selected rosacea data from the IEU Open GWAS project (https://gwas.mrcieu.ac.uk/, accessed on 15 May 2023) (Appendix A).

### 2.3. Selection of Instrumental Variables

The study’s flowchart is illustrated in Figure 1. Concisely, the gut microbiota was employed as the exposure, while dermatological disease was used as the outcome. We selected one threshold for SNPs smaller than the genome-wide statistical significance threshold (5 × 10^−8^) as the IVs. Unfortunately, only a limited number of gut microbiota were selected as IVs following our SNP selection. To explore a wider range of relationships between dermatophytosis and gut microbiota for more comprehensive results, we established a second threshold to identify SNP significance levels smaller than the locus (1 × 10^−5^). We chose a secondary set of IVs to uncover additional potential causal relationships. We set the linkage disequilibrium correlation coefficients to R^2^ < 0.01 and a clumping distance > 5000 kb to ensure no linkage disequilibrium among the chosen IVs. To lessen the effect of weak instrument bias on causal inference, we applied the formula F = β^2^ exposure/SE^2^ exposure to evaluate the potency of the IVs and removed those with F < 10 [12]. Echo SNPs, which are those with identical base pairs on the forward and reverse strands (such as A/T and C/G), can be ambiguous when genotyping. Therefore, we eliminated these SNPs to reduce this potential source of error and thus enhance the accuracy of our analysis.

### 2.4. Statistical Analysis

Currently, several popular MR methods, including the inverse variance weighted (IVW) test [13], MR-PRESSO [14], MR-Egger regression [15], and weighted median estimator [16], are routinely utilized for MR analysis. According to related studies, the IVW method delivers more precise inferences of causality during the analysis. Hence, we chose the IVW method as the main approach for examining causality in the two-sample MR (TSMR) analysis [17]. For attributes that contained only one SNP, where the IVW test was not suitable, we employed the Wald ratio test to estimate causal impacts [18]. We utilized MR-PRESSO and MR-Egger regression tests to keep track of potential horizontal pleiotropy effects. A *p* (intercept) value < 0.05 would suggest the existence of horizontal pleiotropy. The heterogeneity among the chosen SNPs was measured using Cochran’s Q statistic, and a “leave-one-out” analysis was performed to exclude each instrumental SNP in turn to identify any potentially heterogeneous SNPs. We applied this method to ascertain the direction of causality in the MR analysis [19]. Subsequently, we executed a reverse MR analysis, using the same configurations and methods as those in the forward MR. To examine any potential reverse causation effects, we treated the four inflammatory skin diseases as exposure factors and gut flora as the outcome variable. All statistical analyses were conducted using the TwoSampleMR and MR-PRESSO packages in R software (version 4.2.2).

## 3. Results

### 3.1. SNP Selection

At the locus-wide significance level (*p* < 1 × 10^−5^), we identified a total of 2832 SNPs as IVs according to the selection criteria. Detailed information about the selected instrumental variables is presented in Appendix A. At the genome-wide statistical significance thresholds (*p* < 5 × 10^−8^), we identified a total of 30 SNPs as IVs (Appendix A).

### 3.2. Results of the TSMR Analysis (Genome-Wide Statistical Significance Threshold, p < 5 × 10^−8^)

Our analysis revealed a causal correlation between acne and one phylum, one family, and one genus (Figure 2). Wald ratio estimates suggested that the phylum *Actinobacteria* (OR = 0.21, 95% CI: 0.16–0.64, *p* = 1.20 × 10^−3^), the family *Bifidobacteriaceae* (OR = 0.42, 95% CI: 0.25–0.70, *p* = 8.00 × 10^−4^), and the genus *Ruminococcus* torques group (OR = 0.36, 95% CI: 0.13–0.99, *p* = 4.80 × 10^−2^) may play a protective role against acne.

Rosacea was associated causally with one genus (Figure 2). Wald ratio estimates indicated that the genus *Ruminococcus torques group* was potentially linked with rosacea (OR = 4.27, 95% CI: 1.04–17.46, *p* = 4.36 × 10^−2^).

### 3.3. Results of the TSMR Analysis (Gene Locus Range Significance Levels, p < 1 × 10^−5^)

Eczema was found to be causally correlated with two bacterial families and three bacterial genera (Figure 3). According to IVW estimates, the family *Veillonellaceae* (OR = 0.94, 95% CI: 0.88–1.00, *p* = 4.60 × 10^−2^), the genus *Dialister* (OR = 0.89, 95% CI: 0.82–0.96, *p* = 3.30 × 10^−3^), the genus *Family XIII UCG001* (OR = 0.91, 95% CI: 0.83–1.00, *p* = 3.91 × 10^−2^), and *Ruminococcaceae UCG004* (OR = 0.92, 95% CI: 0.86–0.99, *p* = 2.41 × 10^−2^) were associated with a reduced risk of eczema. Conversely, the family *Prevotellaceae* (OR = 1.08, 95% CI: 1.00–1.16, *p* = 3.71 × 10^−2^) was potentially associated with an increased risk of eczema.

Acne was associated causally with one order, five families, and seven genera (Figure 3). IVW estimates showed that the order *Bifidobacteriales* (OR = 0.63, 95% CI: 0.43–0.91, *p* = 1.34 × 10^−2^), family *Bifidobacteriaceae* (OR = 0.63, 95% CI: 0.43–0.91, *p* = 1.34 × 10^−2^), family *Desulfovibrionaceae* (OR = 0.56, 95% CI: 0.33–0.94, *p* = 2.7 × 10^−2^), and family *Lactobacillaceae* (OR = 0.76, 95% CI: 0.60–0.97, *p* = 2.43 × 10^−2^) were potentially protective against acne. Similarly, the genera *Candidatus Soleaferrea* (OR = 0.75, 95% CI: 0.60–0.94, *p* = 1.31 × 10^−2^), *Eubacterium coprostanoligenes* (OR = 0.67, 95% CI: 0.46–0.95, *p* = 2.69 × 10^−2^), *Fusicatenibacter* (OR = 0.71, 95% CI: 0.53–0.94, *p* = 1.84 × 10^−2^), *Lactobacillus* (OR = 0.72, 95% CI: 0.58–0.90, *p* = 4.55 × 10^−3^), and *Ruminococcus torques group* (OR = 0.47, 95% CI: 0.30–0.73, *p* = 8.41 × 10^−4^) were also found to have a protective effect against acne. On the other hand, IVW estimates indicated that the family *Bacteroidaceae* (OR = 1.92, 95% CI: 1.15–3.20, *p* = 1.25 × 10^−2^), family *Clostridiaceae1* (OR = 1.63, 95% CI: 1.04–2.56, *p* = 3.35 × 10^−2^), genus *Allisonella* (OR = 1.42, 95% CI: 1.18–1.70, *p* = 2.16 × 10^−4^), and genus *Bacteroides* (OR = 2.25, 95% CI: 1.48–3.42, *p* = 1.37 × 10^−4^) were potentially associated with an increased risk of acne.

Psoriasis was found to be causally correlated with one phylum and two genera (Figure 3). IVW estimates indicated that the phylum *Bacteroidetes* (OR = 0.81, 95% CI: 0.67–0.98, *p* = 3.30 × 10^−2^) and the genus *Prevotella 9* (OR = 0.87, *p* = 4.47 × 10^−2^) were associated with a reduced risk of psoriasis. However, the genus *Eubacterium fissicatena group* (OR = 1.22, 95% CI: 1.10–1.35, *p* = 1.81 × 10^−4^) was potentially associated with an increased risk of psoriasis.

Rosacea was found to be correlated with one class, one order, and one genus (Figure 3). IVW estimates suggested that the class *Deltaproteobacteria* (OR = 1.55, 95% CI: 1.04–2.30, *p* = 3.10 × 10^−2^) and the order *Desulfovibrionales* (OR = 1.50, 95% CI: 1.00–2.25, *p* = 4.97 × 10^−2^) were potentially associated with an increased risk of rosacea. Conversely, the genus *Butyrivibrio* (OR = 0.81, 95% CI: 0.67–0.99, *p* = 3.77 × 10^−2^) was associated with a reduced risk of rosacea. In the Appendix A, we utilized scatter plots as a powerful data visualization tool to illustrate the relationship between gut flora and causal variables associated with skin diseases (Appendix A). Furthermore, all the significant results are outlined in Table 1, while a comprehensive list of results can be found in Appendix A.

### 3.4. Sensitivity Analyses

In our sensitivity assessments, no pleiotropy was detected in the causal estimates (Appendix A). Specifically, the MR-Egger intercept analysis did not uncover any signs of directional pleiotropy between inflammatory skin diseases and gut microbiota (Appendix A). Additionally, Cochran’s Q statistics revealed no significant heterogeneity (*p* > 0.05) (Appendix A). The leave-one-out assay did not show any outliers, indicating that the results were robust and not dependent on any individual gene variants (Appendix A). The MR Steiger directionality test also showed no abnormalities (Appendix A). In the Appendix A, we include funnel plots to provide a visual representation of the instrument variable heterogeneity in our study. The distribution of points around the causal effect line provides insight into the validity of our instrumental variables. These plots aids in the identification and handling of potential heterogeneity, thereby enhancing the robustness of our Mendelian randomization analysis.

### 3.5. Reverse TSMR Analysis

To evaluate any reverse causation effects, we used inflammatory skin diseases as the exposure and gut microbiota as the outcome. We discovered a bidirectional relationship between psoriasis and both the phylum *Bacteroidetes* and the genus *Prevotella 9* (Table 2). The reverse causation sensitivity analysis did not reveal any abnormalities (Appendix A). In order to better understand the complex interactions between the intestinal flora and various skin diseases, a correlation network was constructed and is illustrated in Figure 4.

## 4. Discussion

This research signifies the first comprehensive effort to assess the causative association between gut microbiota and inflammatory skin conditions from a genetic perspective, using the largest genome-wide meta-analysis of gut microbes conducted by the MiBioGen consortium. The burgeoning body of research has unveiled evidence supporting a “skin–gut axis”, with our identification of 26 such causative relationships based on two-sample Mendelian randomization (MR) analysis.

Our two-sample MR (TSMR) research revealed that the genus *Dialister* serves as a protective factor for eczema, with the genus *Dialister* recognized as a propionic acid producer in the gut [20]. It has been found that individuals with eczema have significantly lower concentrations of SCFAs such as propionic acid than healthy individuals [21]. Propionic acid, an essential SCFA, inhibits inflammation induced by Th2 and Th17 bias in patients with eczema [22]. Thus, we propose that the protective influence of the *Dialister* genus group against eczema may be associated with SCFAs, particularly propionate. *Ruminococcaceae* are the primary SCFA producers in the gut microbiota [23]. *Ruminococcaceae UCG004* demonstrated a protective effect against eczema. In one study, the creation of *AP-L01 microgels, which encapsulate Lactobacillus lactis LI01 in algal pectin (AP), alleviated acute liver injury in mice by boosting SCFA producers and reducing pathogenic microbes, thereby altering the gut microbiota. *Ruminococcaceae UCG004* levels were elevated in the AP-LI01 administration group [24]. The family *XIII UCG001 genus* has been identified as a protective factor against eczema. Some research has found that Family *XIII UCG001* plays a protective role in certain inflammatory conditions, such as scleritis and rheumatoid arthritis [25]. Further exploration is needed to understand the mechanism through which Family *XIII UCG001* functions.

The order *Bifidobacteriales* serves as a protective factor against acne. *Bifidobacterium* is known to enhance the gut mucosal barrier and reduce the levels of lipopolysaccharide (LPS) in the intestine. There may be a certain link between LPS and acne. Studies have shown that LPS endotoxins are present in the blood of acne patients, and these patients exhibit high reactivity to LPS. A study involving 40 acne patients demonstrated the presence of LPS endotoxins in their blood and a high reactivity toward LPS [26]. The family *Bifidobacteriaceae* has been identified as a risk factor for acne. It has been associated with the serum concentrations of various inflammatory markers, including TNF-α and IL-6 [27]. The inflammatory factors IL-1β and TNF-α have a close relationship with the pathogenesis of acne [28]. The genus *Bacteroides* has been identified as a risk factor for acne and has been reported to be associated with various inflammatory conditions [29]. Toxins produced by *Bacteroides* have been shown to trigger the NF-κB and MAPK signaling pathways in activated B cells, leading to the secretion of interleukin-8 (IL-8) and TNF-α [30]. The genus *Candidatus Soleaferrea* is a protective factor against acne. *Candidatus Soleaferrea* is regarded as a beneficial microbiota, although research on it is currently limited. The genus *Fusicatenibacter* also serves as a protective factor against acne. *Fusicatenibacter* has been demonstrated to stimulate the production of the anti-inflammatory cytokine interleukin-10 (IL-10) in gut wall monocytes extracted from ulcerative colitis (UC) models in patients and mice. IL-10 can suppress inflammatory responses [31]. Upon stimulation by *Propionibacterium acnes* (*P. acnes*), the secretion of TNF-α and proinflammatory IL-8 is enhanced in peripheral blood mononuclear cells (PBMCs) of individuals with acne. In contrast, the secretion of anti-inflammatory IL-10 by the PBMCs of acne patients significantly decreases. Furthermore, the capability of CD14 cells in acne patients to engulf P. acnes bacteria is impaired, but the introduction of exogenous IL-10 to PBMC cultures can reinstate phagocytic activity [32]. Therefore, the genus *Fusicatenibacter* may play a protective role in acne through the secretion of IL-10. The genus *Lactobacillus* is a protective factor against acne. A randomized controlled study showed that supplementation with the probiotic *Lactobacillus* can decrease the gene expression of insulin-like growth factor 1 (IGF1) by 32% and increase the gene expression of forkhead box protein O1 (FOXO1) by 65%, improving the appearance of adult acne [33]. Other genera were not found to be mechanistically associated with the presence of acne, and later stages need to be further explored.

The phylum *Bacteroidetes* serves as a protective factor in psoriasis. A review of several studies revealed a decreased abundance of *Bacteroidetes* in psoriasis patients compared to healthy controls [34,35]. Interleukin 17 (IL-17), a proinflammatory cytokine, plays a pivotal role in the pathogenesis of psoriasis. A decrease in regulatory T cell (Treg) levels in psoriasis patients leads to an imbalance between effector T cells and suppressor T cells [36]. Studies on patients with obstructive sleep apnea (OSA) have shown reduced numbers of *Bacteroidetes* to be associated with Th17/Treg imbalance [37]. We propose that *Bacteroidetes* may have a similar role in psoriasis. The phylum *Bacteroidetes* is significant among the short-chain fatty acid-producing microflora in the gut, breaking down dietary fiber and other complex carbohydrates to produce short-chain fatty acids (SCFAs) such as butyric acid and propionic acid [38]. SCFAs exhibit anti-inflammatory characteristics in psoriasis and have the ability to activate regulatory T cells in the colon, aiding in maintaining their equilibrium [39]. The administration of SCFAs can ameliorate psoriasis symptoms by inhibiting histone deacetylase (HDAC) in Tregs and rejuvenating their activity. It has been found to decrease splenomegaly and IL-17 expression and to stimulate IL-10 and Foxp3 in the spleen [40]. The genus *Prevotella 9*, also protective against psoriasis, is part of the *Bacteroidetes* phylum. It exhibits high fermentability of cellulose and other polysaccharides and is adept at producing large amounts of SCFAs [41]. Therefore, we suggest that both *Bacteroidetes* and *Prevotella 9* protect against psoriasis due to their SCFA production. The genus *Eubacterium fissicatena group*, a risk factor for psoriasis, is not yet clearly associated with a specific mechanism in the literature. We discovered a bidirectional relationship between psoriasis and *Bacteroidetes*, as well as *Prevotella 9*. However, this relationship may be influenced by other, uncontrolled confounding factors. This necessitates further study to confirm potential complex interactions.

The class *Deltaproteobacteria* and order *Desulfovibrionales* are both risk factors for rosacea. Both belong to the *Proteobacteria* phylum, with *Desulfovibrionales* being part of the *Deltaproteobacteria* class. This suggests that these bacteria may play an important role in rosacea. In a study on Crohn’s disease, an inflammatory bowel disease, researchers found *Desulfovibrionales* in the intestinal mucosa and submucosal tissues of patients with the disease, while the bacteria were not detected in healthy individuals. This suggests that *Desulfovibrionales* may contribute to the pathology of inflammatory bowel disease [42]. Given that rosacea often coexists with inflammatory bowel diseases such as ulcerative colitis and Crohn’s disease [43], the mechanisms related to these two microbiota in rosacea need to be further explored. The genus *Ruminococcus torques group* has been identified as a potential risk factor for rosacea. Research has indicated a positive correlation between this bacterial group and inflammatory markers such as IL-1β, IL-6, IFN-γ, and TNF-α [44]. These factors have been found to play a crucial role in the pathophysiology of rosacea [28,45].

In order to delve into the potential impact of specific substances within the gut microbiome on the development and treatment of skin diseases, utilizing artificial neural network technology to simulate and predict the generation of microbial secreted substances might be an effective approach [46]. After initial conclusions are drawn from the simulations, these predictions can be validated through in vitro or in vivo experiments. By thoroughly studying these molecular mechanisms, we can better understand how the gut microbiome influences the development and severity of inflammatory skin diseases. This understanding not only provides a theoretical foundation for aiding the treatment of inflammatory skin diseases by modulating the gut microbiome in the future but also helps us identify possible new drug targets, thereby developing new treatment methods. Ultimately, through this interdisciplinary research approach, we aim to provide new possibilities for controlling the symptoms of inflammatory skin diseases and improving the quality of life of patients.

This study has several strengths. First, it is the pioneer in employing bidirectional two-sample Mendelian randomization (TSMR) to elucidate a causal relationship between gut microbiota and inflammatory skin diseases. This relationship is not confounded by potential confounding factors or subjected to reverse causality. Second, we assessed instrumental variables in each group to ensure that no bias was introduced. Third, our findings were robust and validated through multiple analyses with consistent results. In addition, our results were verified for reliability via comprehensive sensitivity analyses. These findings add significantly to our understanding of the role of gut microbiota in skin health and disease, highlighting the importance of microbial balance in maintaining skin health. They also point toward the potential for future research to focus on developing microbiome-based therapies for the treatment of these skin conditions. Therefore, further studies are required to validate these relationships and to explore the mechanisms underpinning these links.

However, certain limitations should be noted. First, the bulk of the patient data in our GWAS summary were of European origin, with a limited amount of gut microbiota data coming from other ethnic groups. This could potentially introduce a bias in our estimates and limit the universal applicability of our results. Secondly, due to the limited number of instrumental variables included in the gut microbiota GWAS data and the lack of species-level data, it was not possible to ascertain if there was an overlap of participants in the GWAS data for the exposure and outcome variables. Additionally, given the extensive number and complex hierarchical structure of microbial taxa, adjustments for multiple comparisons, particularly global multiple corrections, might be overly conservative, potentially leading to an increased risk of Type II errors (false negatives). As such, negative results should be interpreted with caution, as they do not definitively rule out a causal relationship.

## 5. Conclusions

In conclusion, we conducted a comprehensive assessment of the causal relationships between gut microbiota and an array of inflammatory skin diseases. We found that acne was associated with two bacterial families and three genera, revealing one positive and four negative causal pathways. Furthermore, acne exhibited connections with one phylum, one order, five families, and seven genera, encompassing four positive and ten negative causal directions. Psoriasis demonstrated a correlation with one phylum and two genera, with one positive and two negative causal pathways identified. Rosacea was found to be associated with one bacterial class, one order, and two genera, indicating three positive and one negative causal connections. These findings underscore the complex interplay between gut microbiota and inflammatory skin conditions and highlight the potential of microbiota modulation as a novel therapeutic avenue. Further research is warranted to validate these associations and to elucidate the underlying molecular mechanisms. This will ultimately pave the way toward personalized microbiota-based interventions for the prevention and treatment of these common and debilitating skin disorders.

## Figures and Tables

**Figure 1 microorganisms-11-02586-f001:**
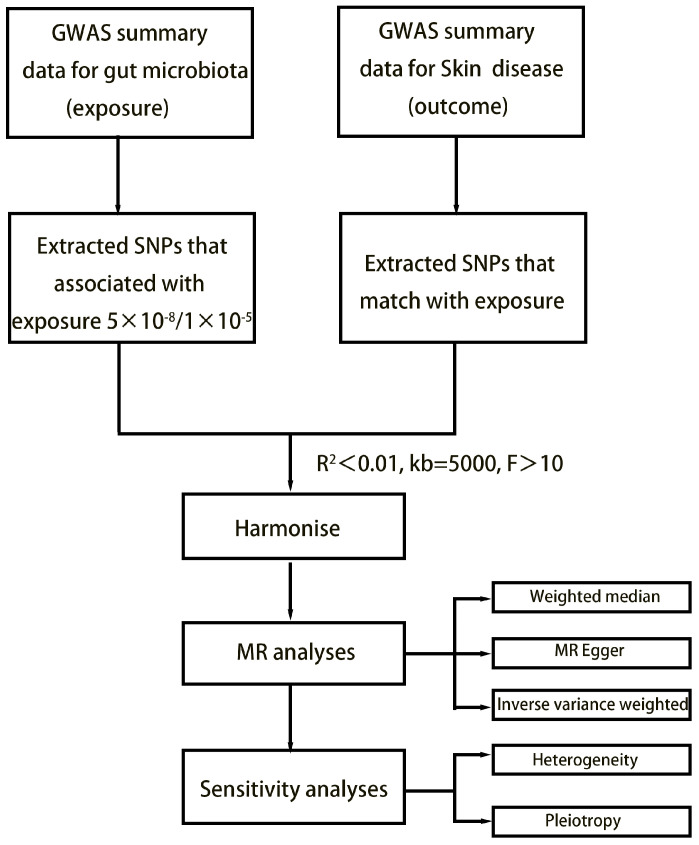
Flowchart of MR analysis.

**Figure 2 microorganisms-11-02586-f002:**
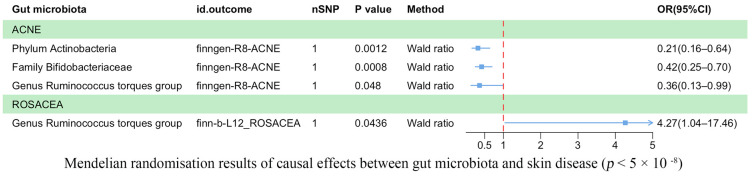
Mendelian randomization results on the causal relationship between the gut microbiome and the risk of four inflammatory skin diseases (*p* < 5 × 10^−8^).

**Figure 3 microorganisms-11-02586-f003:**
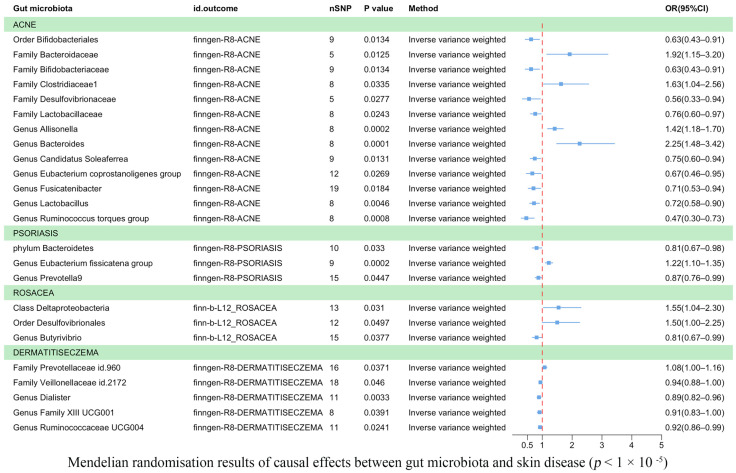
Mendelian randomization results on the causal relationship between the gut microbiome and the risk of four inflammatory skin diseases (*p* < 1 × 10^−5^).

**Figure 4 microorganisms-11-02586-f004:**
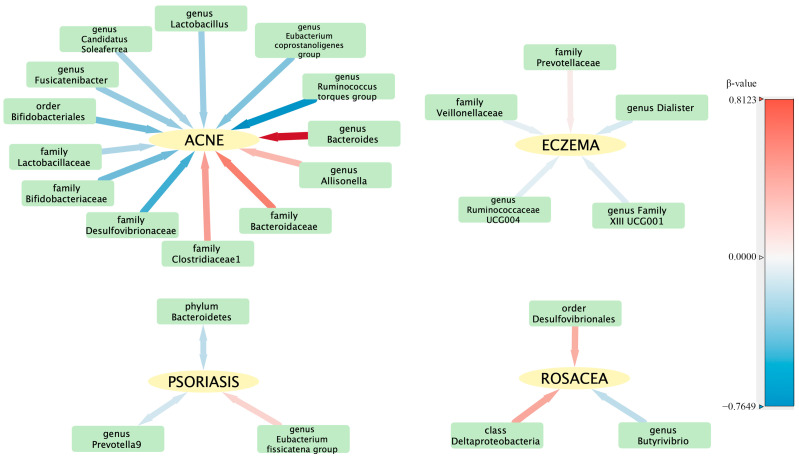
Visualization of the correlation network between intestinal flora and skin diseases. The hue of each line signifies the magnitude of the β-value, while the thickness of each line represents its absolute value, illustrating the strength and direction of the relationship.

**Table 1 microorganisms-11-02586-t001:** Mendelian randomization (MR) results of causal effects between gut microbiome and four kinds of skin diseases.

Threshold/Method	Outcome	Id. Exposure	Id. Outcome	Nsnp	Beta	Se	*p*-Value	Trait
5 × 10^−8^/Wald ratio	ACNE	ebi-a-GCST90017110	finngen-R8-ACNE	1	−1.14272	0.351571	0.001153	Phylum *Actinobacteria* id.400
1 × 10^−5^/IVW	ACNE	ebi-a-GCST90017093	finngen-R8-ACNE	9	−0.46978	0.190061	0.013446	Order *Bifidobacteriales* id.432
5 × 10^−8^/Wald ratio	ACNE	ebi-a-GCST90016929	finngen-R8-ACNE	1	−0.875	0.26169	0.000827	Family *Bifidobacteriaceae* id.433
1 × 10^−5^/IVW	ACNE	ebi-a-GCST90016927	finngen-R8-ACNE	5	0.651397	0.260806	0.012503	Family *Bacteroidaceae* id.917
1 × 10^−5^/IVW	ACNE	ebi-a-GCST90016929	finngen-R8-ACNE	9	−0.46978	0.190061	0.013446	Family *Bifidobacteriaceae* id.433
1 × 10^−5^/IVW	ACNE	ebi-a-GCST90016931	finngen-R8-ACNE	8	0.488064	0.229627	0.033548	Family *Clostridiaceae1* id.1869
1 × 10^−5^/IVW	ACNE	ebi-a-GCST90016935	finngen-R8-ACNE	5	−0.58438	0.265392	0.027668	Family *Desulfovibrionaceae* id.3169
1 × 10^−5^/IVW	ACNE	ebi-a-GCST90016941	finngen-R8-ACNE	8	−0.26929	0.119557	0.024295	Family *Lactobacillaceae* id.1836
5 × 10^−8^/Wald ratio	ACNE	ebi-a-GCST90017066	finngen-R8-ACNE	1	−1.0184	0.515063	0.048014	Genus *Ruminococcus torques group* id.14377
1 × 10^−5^/IVW	ACNE	ebi-a-GCST90016963	finngen-R8-ACNE	8	0.347787	0.094021	0.000216	Genus *Allisonella* id.2174
1 × 10^−5^/IVW	ACNE	ebi-a-GCST90016968	finngen-R8-ACNE	8	0.812298	0.213034	0.000137	Genus *Bacteroides* id.918
1 × 10^−5^/IVW	ACNE	ebi-a-GCST90016976	finngen-R8-ACNE	9	−0.28914	0.116587	0.013137	Genus *Candidatus Soleaferrea* id.11350
1 × 10^−5^/IVW	ACNE	ebi-a-GCST90016997	finngen-R8-ACNE	12	−0.40734	0.184114	0.026938	Genus *Eubacterium coprostanoligenes* group id.11375
1 × 10^−5^/IVW	ACNE	ebi-a-GCST90017011	finngen-R8-ACNE	19	−0.34742	0.147413	0.018436	Genus *Fusicatenibacter* id.11305
1 × 10^−5^/IVW	ACNE	ebi-a-GCST90017030	finngen-R8-ACNE	8	−0.32463	0.11442	0.004552	Genus *Lactobacillus* id.1837
1 × 10^−5^/IVW	ACNE	ebi-a-GCST90017066	finngen-R8-ACNE	8	−0.76493	0.229111	0.000842	Genus *Ruminococcus torques group* id.14377
1 × 10^−5^/IVW	PSORIASIS	ebi-a-GCST90017111	finngen-R8-PSORIASIS	10	−0.21072	0.098831	0.032997	Phylum *Bacteroidetes* id.905
1 × 10^−5^/IVW	PSORIASIS	ebi-a-GCST90016999	finngen-R8-PSORIASIS	9	0.197321	0.052692	0.000181	Genus *Eubacterium fissicatena group* id.14373
1 × 10^−5^/IVW	PSORIASIS	ebi-a-GCST90017045	finngen-R8-PSORIASIS	15	−0.13764	0.068571	0.044716	Genus *Prevotella 9* id.11183
1 × 10^−5^/IVW	ROSACEA	ebi-a-GCST90016915	finn-b-L12_ROSACEA	13	0.435394	0.201843	0.030998	Class *Deltaproteobacteria* id.3087
1 × 10^−5^/IVW	ROSACEA	ebi-a-GCST90017097	finn-b-L12_ROSACEA	12	0.405676	0.2067	0.049689	Order *Desulfovibrionales* id.3156
5 × 10^−8^/Wald ratio	ROSACEA	ebi-a-GCST90017066	finn-b-L12_ROSACEA	1	1.450923	0.718926	0.043572	Genus *Ruminococcus torques group* id.14377
1 × 10^−5^/IVW	ROSACEA	ebi-a-GCST90016975	finn-b-L12_ROSACEA	15	−0.20966	0.100904	0.037723	Genus *Butyrivibrio* id.1993
1 × 10^−5^/IVW	DERMATITISECZEMA	ebi-a-GCST90016948	finngen-R8-DERMATITISECZEMA	16	0.074607	0.035786	0.037084	Family *Prevotellaceae* id.960
1 × 10^−5^/IVW	DERMATITISECZEMA	ebi-a-GCST90016956	finngen-R8-DERMATITISECZEMA	18	−0.06238	0.031264	0.046024	Family *Veillonellaceae* id.2172
1 × 10^−5^/IVW	DERMATITISECZEMA	ebi-a-GCST90016988	finngen-R8-DERMATITISECZEMA	11	−0.11711	0.039833	0.003282	Genus *Dialister* id.2183
1 × 10^−5^/IVW	DERMATITISECZEMA	ebi-a-GCST90017009	finngen-R8-DERMATITISECZEMA	8	−0.09576	0.046417	0.039115	Genus *Family XIII UCG001* id.11294
1 × 10^−5^/IVW	DERMATITISECZEMA	ebi-a-GCST90017055	finngen-R8-DERMATITISECZEMA	11	−0.07922	0.035124	0.024099	Genus *Ruminococcaceae UCG004* id.11362

**Table 2 microorganisms-11-02586-t002:** Bidirectional MR results of the causal effects between gut microbiome and skin disease.

Id. Outcome	Gut Microbiota (Outcome)	Exposure	Method	Number of Snps	Beta	Se	*p*-Value	Correct_Causal Direction	Steiger_Pval
ebi-a-GCST90017045	genus *Prevotella 9*	PSORIASIS	IVW	75	0.04	0.02	0.02	TRUE	0.001
ebi-a-GCST90017111	phylum *Bacteroidetes*	PSORIASIS	IVW	80	0.04	0.01	0.002	TRUE	0.0002

## Data Availability

The summary data for FINNGEN are readily accessible for download from the FinnGen version 8 website: https://r8.risteys.finngen.fi/ (accessed on 15 May 2023). Similarly, the summary data for IEU can be obtained from the official website: https://gwas.mrcieu.ac.uk/ (accessed on 15 May 2023). Other datasets generated and/or examined in the course of the present study are also publicly accessible. They are included within this published article and in its Appendix A.

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
