# Peer review of "Exploring the Association between Gut Microbiota and Inflammatory Skin Diseases: A Two-Sample Mendelian Randomization Analysis"

_microorganisms, 2023, doi:10.3390/microorganisms11102586_

Round 1

Reviewer 1 Report

Although the authors’ findings that connections between the gut microbiome and inflammatory skin conditions using the Mendelian randomization (MR) are very interesting, numbers of points need clarifying and certain statements require further justification. These are given below.

<Points>

1.      Although the study was used only human intestinal microbiome data, the authors did not describe Ethics Statement. The authors should describe so-called ethics committee approval number(s) and date clearly.

2.      The authors’ affiliations “123” should be changed to “1, 2, 3”.

3.      “5 x 10 -8” (line 111) and “1 x 10 -5” (line 114) should be changed to 5 x 10 <sup>-8</sup>” and “1 x 10 <sup>-5</sip>”, respectively.

4.      In Table 1, “Thresh” and “Method” should be described in one line. In addition, “Thresh” may be “Threshold”.

5.      In Figures 2 and 3, some letters are too small to see.

6.      “Published 2020 June 12” in Ref. 34 should be deleted.

Author Response

Response to point 1:

Thank you for your valuable insights and thorough review of our research. We fully understand the importance of ethical approval and agree that it should be clearly stated in studies involving human participants. Our study primarily utilized data from the public domain, particularly from previously published research and publicly available statistical data. We did not collect any new data directly from human participants, nor did we encounter any personally identifiable information. All the original studies we analyzed had received the necessary approvals from their respective ethics committees, and participants had provided informed consent.

Our aim is to explore the association between the human gut microbiome and inflammatory skin diseases using the existing public data, eliminating the need for any new data collection. We are confident that our research adheres to all relevant ethical guidelines and standards.

To ensure full transparency and ethical compliance, we are willing to add an "Ethics approval and consent to participate" section at the end of our paper. In this section, we will elucidate how we ensured ethical compliance with all data utilized, and how we adhered to all relevant ethical standards and guidelines.

We are very open to continuing this discussion and are willing to make any necessary amendments to ensure our research meets all ethical and editorial requirements. Please feel free to contact us for further discussion or clarification. We look forward to your feedback and appreciate your efforts in enhancing the quality of our manuscript.

Here is the original text added:

Ethics Approval and Consent to Participate:This study is based on publicly available data. All original data have received the necessary approval from the relevant ethics committees, and all participants have provided informed consent. Our research did not involve any new data collection or personal identification information, thus no new ethical review was conducted. We adhered to all relevant ethical standards and guidelines, ensuring transparency and ethical responsibility in data handling.

Response to point 2, 3 and 6:

We sincerely appreciate your meticulous attention to detail and your valuable feedback on our manuscript. Your suggestions have significantly contributed to enhancing the quality and accuracy of our paper. Here are the amendments we have made based on your recommendations:

  1. We have amended the authors' affiliations from "123" to “1, 2, 3” as advised.
  2. As suggested, the expressions "5 x 10 -8" (line 111) and “1 x 10 -5” (line 114) have been revised to "5 x 10<sup>-8</sup>" and "1 x 10<sup>-5</sup>", respectively.
  3. We have removed the publication date "Published 2020 June 12" from Reference 34 as directed.

Response to point 4:

Thank you for your meticulous review and valuable suggestions on our manuscript. Following your guidance, we have revised Table 1. The term "Thresh" has been corrected to "Threshold," and we have combined the "Threshold" and "Method" columns into one line. We hope that these revisions meet your expectations and make our manuscript more accurate and clear.

Response to point 5:

Thank you for bringing to our attention the readability issue concerning Figures 2 and 3. We have taken your feedback into account and have revised the figures accordingly. The font size of the text within Figures 2 and 3 has been increased to ensure better visibility and readability.

Reviewer 2 Report

Dear authors,

Really interesting in silico research article. Gut microbio is indeed linked (if not fully associated) to inflammatory-state diseases (and not only skin diseases) The article is concise, well presented and well written. The findings are promising. Limitations have been mentioned; the topic and its pertinence and originality of the study merits the consideration of the article for publication. 

Nevertheless, it will be important (in a next step) to perform validations studies; Also, it is important to describe the skin disorders you have selected and suggest potential molecular mechanisms induced by the microbe to cause such a skin disease. Eventually, I strongly suggest to refer amd cite the following articles:

1-https://www.frontiersin.org/articles/10.3389/fimmu.2020.580208/full

2-Saber WIA, et al. A comparative study using response surface methodology and artificial neural network towards optimized production of melanin by Aureobasidium pullulans AKW. Sci Rep. 2023 Aug 19;13(1):13545. doi: 10.1038/s41598-023-40549-z. PMID: 37598271; PMCID: PMC10439932.

3-https://journals.ansfoundation.org/index.php/jans/article/view/4540

best,

The reviewer

Fine 

Author Response

Thank you very much for your insightful comments and constructive feedback on our manuscript. We sincerely appreciate the time and effort you have invested in reviewing our work. Your suggestions have provided us with a valuable perspective that has significantly enhanced the quality and clarity of our study.

In light of your feedback, we have meticulously revised our manuscript, particularly the discussion section where we have incorporated references to the articles you have recommended. We have added the following text to further elucidate the prospective impact of gut microbiome substances on skin diseases, and the potential of artificial neural network technology in exploring these dynamics:

"In order to delve into the potential impact of specific substances within the gut microbiome on the development and treatment of skin diseases, utilizing artificial neural network technology to simulate and predict the generation of microbial secreted substances might be an effective approach [46]. After initial conclusions are drawn from the simulations, these predictions can be validated through in vitro or in vivo experiments. By thoroughly studying these molecular mechanisms, we can better understand how the gut microbiome influences the development and severity of inflammatory skin diseases. This understanding not only provides a theoretical foundation for aiding the treatment of inflammatory skin diseases by modulating the gut microbiome in the future but also helps us identify possible new drug targets, thereby developing new treatment methods. Ultimately, through this interdisciplinary research approach, we aim to provide new possibilities for controlling the symptoms of inflammatory skin diseases and improving the quality of life of patients."

Round 2

Reviewer 1 Report

Most of points were suitably revised in microorganism-2638785v2. Several points to be changed to improve the work.

<Points>

1.      In Author Contributions (line 399), who in “WYZ” should be clarified. In the first page, there is no “WYZ”.

2.      In Author Contributions (lines 397–399), “LJ”, “GJ”, “YJ”, “CP”, “DY”, “LY”, and “WM” may be “Junhao Long”, “Jinglan Gu”, “Juexi Yang”, “Pu Chen”, “Yan Dai”, “Yun Lin” and “Ming Wu”, respectively. If so, they should be changed to “JL”, “JG”, “JY”, “PC”, “YD”, “YL”, and “MW”, respectively.